# Beyond the Rosetta Stone: Unification Forces in Generalization Dynamics

## Abstract

Large language models (LLMs) struggle with cross-lingual knowledge transfer: they hallucinate when asked in one language about facts expressed in a different language during training. This work introduces a controlled setting to study the causes and training dynamics of this phenomenon by training small Transformer models from scratch on synthetic multilingual datasets. We identify a learning phase wherein a model develops either separate or unified representations of the same facts across languages, and show that unification is essential for cross-lingual transfer. We demonstrate that the degree of unification depends on fact-language correlation (mutual information) and the ease of language identification early in pre-training. Based on these insights, we propose a unifying perspective explaining a range of prior observations concerning cross-lingual transfer in multilingual LLM made. Our work shows how controlled settings can shed light on pre-training dynamics and suggests new directions for improving cross-lingual transfer in LLMs.

## 1 Introduction

Language models hallucinate facts. This has been attributed to training and sampling noise, gaps in pretraining data (Xu et al., 2024), and misaligned incentives in post-training (Schulman, 2023). However, these fail to explain *cross-lingual* factual errors: cases where models accurately answer questions when posed in the same language as the training data, yet hallucinate when prompted in a different (often lower-resource) language (Goldman et al., 2025). Failures of cross-lingual transfer exacerbate disadvantages faced by speakers of underrepresented languages, and increasing model scale does not solve the problem (Aggarwal et al., 2025; Qi et al., 2023). LLMs have been found to develop both a lingua franca for factual knowledge (typically based on English) and distinct language silos (Aggarwal et al., 2025; Lim et al., 2025b; Schut et al., 2025; Lim et al., 2025a, inter alia), and their hidden representations can be language-agnostic or language-specific depending on the layer (Wang et al., 2025). However the root cause of these phenomena is not understood, as most research on cross-lingual transfer analyzes models as static artifacts. Such analysis, while valuable, cannot explain how knowledge *arises* during training, and therefore cannot lead to effective pre-training interventions. While some have investigated the training dynamics of knowledge acquisition in multilingual LLMs (Zeng et al., 2025; Liu et al., 2025), their approach is non-interventional and does not establish a causal link between data properties and cross-lingual transfer. In this work, we study what causes cross-lingual hallucinations, and how to mitigate them. We use a "Petri dish" methodology, training small transformer models from scratch on synthetic datasets and systematically varying their distributional properties. This setup allows us to analyze models' learning dynamics during pre-training. In particular, we identify a crucial early phase where a model develops either unified or separate representations for identical facts across languages, and find that the degree of representational unification, computed over *training* examples, is predictive of cross-lingual generalization in the fully trained model.

Our study reveals two primary causes of unification. First, expressing the same information in different languages facilitates the development of shared cross-lingual representations. This builds on findings from monolingual research (Allen-Zhu, 2024) where including multiple paraphrases of the same fact in training was found to improve recall. Second, and more surprisingly, we find that the distributional properties of the *monolingual* (non-parallel) portion of the dataset can induce

representational separation. Namely, separation occurs when the language of an example is both easy to extract, and is itself a useful prior for predicting the response distribution.

In summary, our core contributions are:

1. We introduce a Petri dish setup in which same-language generalization is reliably observed, while cross-lingual transfer can be independently modulated (Sec. 3).

2. We analyze pre-training dynamics and identify a crucial early phase where a model develops either unified or separate representations (Sec. 4).

3. We introduce a metric to characterize unification of representations across languages that is strongly predictive of cross-lingual knowledge transfer (Sec. 5).

4. We show that cross-lingual transfer can be improved without increasing the amount of multi-lingual data, but rather by changing properties of the monolingual data such that the model takes longer to learn the language feature (Sec. 6).

5. Beyond our synthetic setting, our findings provide a unifying perspective on seemingly disparate observations from prior work about cross-lingual transfer in LLMs, including the roles of script, vocabulary size, and embeddings (Sec. 2).

Finally, we note that our Petri dish model of cross-lingual knowledge transfer could be interpreted more generally as a model of transfer across semantic paraphrases. Thus, we believe our study can have applications beyond factual recall - we discuss these in Sec. 7.

## 2 INTERPRETING EXISTING OBSERVATIONS

Our key contribution is to explain why language models might fail to transfer factual knowledge across languages. Specifically, we create a petri dish environment whose emergent properties naturally demonstrate that if language identity is easy to extract early in training, its representational footprint grows faster than that of the true, language-independent facts (Lampinen et al., 2024), creating 'language silos' that block transfer (Lim et al., 2025b). This lens of *language feature extractability* explains various seemingly unrelated observations in prior work:

**Script similarity**   Several studies have observed that cross-lingual performance correlates more strongly with script similarity than with geographical, linguistic, or cultural proximity. For instance, Greek—an Indo-European language—shows low correlation with other European languages (Liu et al., 2025). Conversely, Indonesian—an Austronesian language utilizing the Latin script—demonstrates better cross-lingual transfer with English than other Asian languages (Goldman et al., 2025).

While these findings may seem counter-intuitive, our framework offers a clear explanation: language identity is trivial to extract from a unique script (e.g. Greek). Consequently, the "language feature" is absorbed early in training, leading to siloed representations. In contrast, shared scripts (like Latin) delay this extraction, allowing for better transfer.

**Shared vocabulary**   Patil et al. (2022) propose increasing vocabulary overlap between languages prior to training, demonstrating that this improves cross-lingual knowledge transfer. We interpret this result similarly: increased vocabulary overlap makes the language identity feature harder to extract, thereby reducing its footprint in the model's representations.

Furthermore, Qi et al. (2023) find that vocabulary overlap correlates strongly with cross-lingual consistency. They thus speculate that larger models do not necessarily show better cross-lingual consistency because their vocabularies are larger. We add another layer of explanation: larger vocabularies make language identification easier, which in turn increases the prominence of spurious language features. As the original analysis was performed on older models, we re-run it with five recent LLMs (Gemma-2, Gemma-3, Llama 3, Qwen 3, and Mistral. See App. A.15.6) and on a recent benchmark of cross-lingual factual knowledge transfer (Goldman et al., 2025). In line with prior findings, we observe a high (0.69) Pearson correlation between the model-specific vocabulary overlap between languages and the degree of cross-lingual transfer.

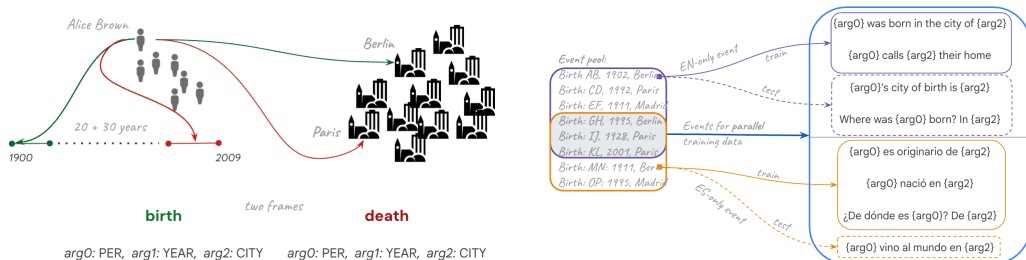

(a) KG is populated with events of two types      (b) Templates to express birth-city facts in EN and ES

Figure 1: Birth and death events are created for every entity by sampling from a set of years (disjoint) and cities (same pool). A dataset is comprised of monolingual (expressed in either EN or ES) and parallel (expressed in both languages) events. Arrows point from a particular event to the training templates (solid line) or in-language test templates (dashed). All verbalizations in the other language are part of the cross-lingual test set. To simplify, only birth templates with *arg0, arg2* are shown.

**Aligning embeddings** Our work also provides conceptual underpinnings for other works such as Li et al. (2024), who initialize embeddings for aligned words to be similar prior to pretraining to promote shared representations. This pre-alignment operates by encouraging unified representations that we show are critical to cross-lingual transfer, while also reducing the extractability of the language signal we discuss above.

## 3 CROSS-LINGUAL FACTUAL RECALL IN A PETRI DISH

In this section we describe our Petri dish methodology, from creating pre-training data to training and evaluating models. First, we create a synthetic knowledge graph by generating language-agnostic events (e.g., *birth, death*). From a single event we derive multiple **facts**, each of which has **subject** and **attribute** arguments. For example, it may be a fact that `Alice Brown` (subject) was born in `Berlin` (birth-place attribute), or that `Alice Brown` was born in the year `1902` (birth-year attribute) (see Fig. 1a and App. A.1 for more details). Once built, the KG is frozen and its events serve as the basis for multiple experiments, so the training datasets express the same set of information.

**Synthetic Languages** We develop synthetic languages to express the KG. Each language is defined by a set of templates, where each template corresponds to a KG fact type (e.g., `birth-year`) and includes slots for its arguments (Fig. 1b). All experiments use two languages and no tokens are shared between templates (unless stated otherwise). See App. A.3 and A.1 for further details

**KG to a pre-training dataset** Some events are expressed in a single language (*non-parallel data*), others in *both* languages. The latter, cross-lingual events, are verbalized with *every* template in the training set and comprise its *parallel data* (events within the overlap in Fig. 1b). Events in the non-parallel data are still verbalized using multiple templates from the same language. Note that we feed the model with individual examples, hence our use of the word *parallel* only means that the same event is encountered in both languages at some point during training, not within the same sequence. Intuitively, increasing the amount of parallel data should improve generalization across languages. To measure this effect, we vary the proportion of cross-lingual events and generate multiple datasets for different ratios.

**Measuring In-language and Cross-language Generalization** The task of factual recall is to retrieve the correct attribute when presented with a statement truncated before its final argument (e.g., given *The year of Alice Brown's birth is*, the model must retrieve *1902*). In evaluating factual recall, we must distinguish between mere string *memorization* and genuine *generalization*. Following Allen-Zhu (2024), we assess generalization by reserving at least one verbalization for each monolingual fact exclusively for evaluation (Fig. 1b). This setup allows us to measure both **in-language** generalization (using held-out verbalizations in that same language) and **cross-lingual** generalization. In addition to the overall in- (or cross-) language accuracy for some experiments we report accuracy per **fact type** – for *birth-year, birth-city, death-year, death-city*.

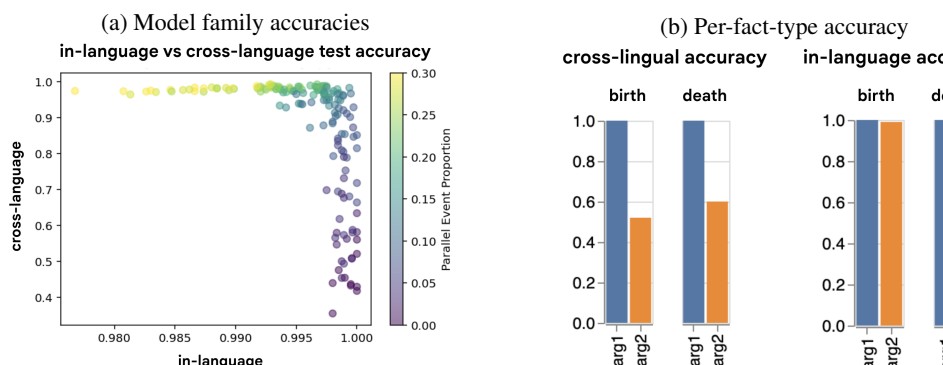

Figure 2: **Left**: In-language versus cross-language test label probability across Pythia models (pretrained on datasets expressing the same facts in the same languages) across parallel data ratios. In-language performance does not predict cross-lingual performance. **Right**: Accuracies for in-language and cross-language evaluation of a particular model for the four fact types. Model attains perfect cross-lingual accuracy when predicting years (*arg1*) but not cities (*arg2*), while attaining perfect in-language accuracy for that task.

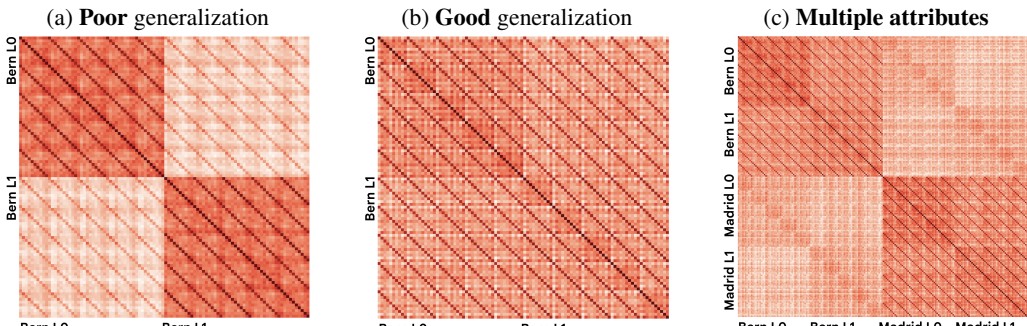

Figure 3: Pairwise cosine similarities between activation-based representations of examples from a model with poor (left), and good (middle) cross-lingual generalization. Examples are taken from the parallel portion of the training data. The left and middle plots visualize 100 examples with the same predicted birth-city attribute (*Bern*) grouped by language (*L0* and *L1*). The right plot visualizes examples with either *Bern* or *Madrid* as the predicted birth-city attribute, also grouped by language within each attribute, from a model with median generalization performance.

**Tiny model training and evaluation**   We train small Transformer models using standard configurations (Pythia, Gemma 2) from random initialization on our synthetic datasets for multiple epochs, monitoring the in- and cross-language accuracies during training. The total parameter count in our models is typically around 2M (six layers, four attention heads, and a hidden size of 128). As described previously, we vary the following parameters when creating a training set: (1) amount of parallel data, (2) the discrepancy in entity frequency between languages. In our experiments we report the fraction of parallel data by event, starting from 0% (no parallel data) and increasing the amount to 30% with a step size of 2%.

As illustrated in Figure 2a, cross-lingual performance cannot be predicted from in-language factual recall. Most of our models achieve nearly perfect in-language generalization, correctly predicting attributes for queries unseen during training. At the same time, cross-lingual generalization ability ranges from 40% to 100%. Figure 2a also confirms that the amount of parallel data is highly predictive of cross-lingual generalization.

## 4   LEARNING STAGES

We analyze representations based on: (1) activations from the residual stream, and (2) gradients of model parameters. By monitoring cosine similarity between representations during training, we can pinpoint when semantically related inputs across languages converge and diverge. To obtain activation-based representations we take the residual stream contents for the token immediately preceding the attribute to be predicted (e.g. *Alice Brown was born **in***). Unless otherwise noted, we

Figure 4: Pairwise similarity matrices between activation-based representations at the token preceding the attribute across checkpoints. Every image pair contrasts a model trained with 8% (left) versus 30% (right) cross-lingual events–the former has poor cross-lingual generalization while the latter generalizes perfectly. Red means high similarity.

(a) [Checkpoint-282] At first, examples of the same **attribute type** (e.g., `city`) are unified (regardless of whether they pertain to `birth` vs. `death`).

(b) [Checkpoint-564] `Birth` vs. `death` attributes diverge. A **checkerboard pattern** emerges within the high-similarity blocks of the poorly generalizing model (left), signaling **separation by language**.

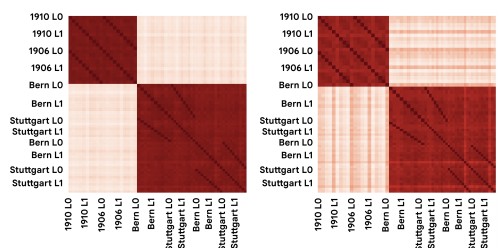

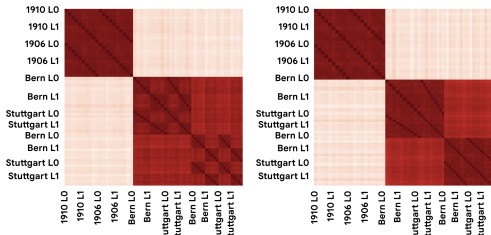

(c) [Checkpoint-1,100] Checkerboarding intensifies (left), e.g. representations for the `death-city` *Bern* in *language-0* are more similar to those for *Stuttgart* in the same language than to those for *Bern* in *language-1*. The opposite trend emerges for the successful model (right).

(d) [Checkpoint-14,000] At later checkpoints, the patterns are stark. The successful model (right) has unified the representations for examples with the same attribute value (e.g., `birth-city-Bern`), while undesired language checkerboarding remains prominent in the other model (left).

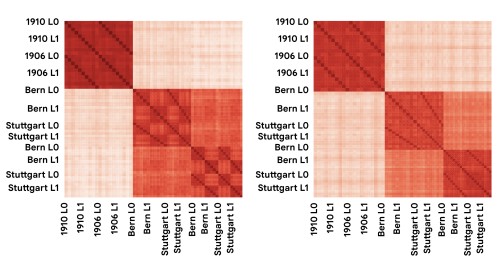

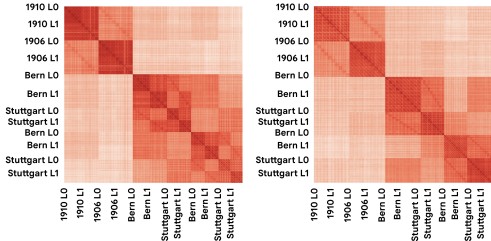

concatenate the activations of each layer to form this representation. The use of such embeddings for the purpose of model analysis and visualization is well established (nostalgebraist, 2020; Ghandeharioun et al., 2024). To obtain gradient-based representations we use the model weights' gradients at the attribute token. Intuitively, if the gradients between two examples are similar, then they exert a similar influence during training, and are processed by similar model parameters. We draw inspiration from methods for identifying influential training inputs by comparing the gradients of training data to test data (Koh & Liang, 2017; Schioppa et al., 2023; Ruis et al., 2025), and leverage a recent, computationally efficient approximation developed by Chang et al. (2025).

Figure 3c shows pairwise cosine similarities between activation-based representations[1] of a Gemma model trained on a 50-50 language split and 16% cross-lingual events. Representations are computed from training examples and, while referring to distinct *birth-year* facts, all have *Bern* or *Madrid* as the (correctly predicted) birth-city attribute. Within each attribute, examples are grouped by language (*L0, L1*). In this model with median generalization ability, examples with the same attribute (e.g., *Madrid*) are more similar to each other than to examples with a different attribute (*Bern*)—this is clearly visible in the two large red blocks. However, within the red matrix corresponding to the same attribute (e.g., *Madrid-Madrid*, bottom right), two sub-blocks are visible, indicating that examples of the same language are more similar than examples across languages. For comparison, plots in Fig. 3a-3b show the similarities for a *single* birth-city attribute (*Bern*), again grouped by language, from two models—with worse and better cross-lingual factual recall (8% vs. 30% cross-lingual events). Only Fig. 3a has four distinct blocks, indicating cross-lingual dissimilarity.

When does separation by language happen in training? Consider Fig. 4, which shows pairwise similarity matrices for two models across checkpoints. One model (on the right in every pair of matrices) eventually achieves perfect cross-lingual transfer while the other (on the left) does not. The models' training sets are identical with respect to factual content, languages used, language

---

[1]Gradient-based plots are in Fig. 17 in the Appendix.

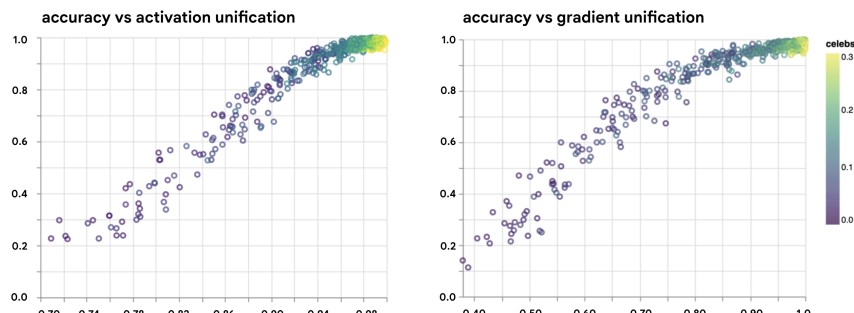

Figure 5: Activation unification scores (left) and gradient unification scores (right) correlate strongly with cross-lingual factual recall accuracy (0.97 and 0.94 PCC, respectively) (see Figure 13 for layerwise results). Each datapoint corresponds to a different model training run. Note that the fraction of celebrity events (denoted by color) also correlates strongly with generalization ability.

split (50-50) and number of examples, but differ in the **amount of parallel data**—30% events vs. 8%. Note that the trends on display are observed consistently across dozens of models trained with different languages, language proportions, model scales, etc. Each matrix in Figure 4 shows pairwise similarities between 300 examples equally split between `birth-year`, `birth-city`, `death-city`, each corresponding to two values (*1906, 1910* for `birth-year`, *Bern, Stuttgart* for `birth-city`, *Bern, Stuttgart* for `death-city`). Again, within each attribute value, examples are grouped by language (*L0* and *L1*). We observe that poorly-generalizing models undergo a signature phase (around checkpoint-564) wherein language identity, rather than semantic equivalence, drives representational similarity (see captions for more details).

We quantify our observations with the concept of a unification score, which predicts cross-lingual generalization. Intuitively, the unification score measures how much the model *avoids* the checkerboarding on the right side of Fig. 4.

## 5 UNIFICATION PREDICTS CROSS-LINGUAL PERFORMANCE

Concretely, the unification score captures the similarity between semantically equivalent cross-lingual datapoints against a baseline of similarity between semantically distinct same-language datapoints. We define it as $Unification(\theta, \mathcal{D}) := E_{X,Y \sim Facts(\mathcal{D})}\big[sim_\theta(X,Y)/sim_\theta(X,X)\big]$ where $X, Y \in Facts(\mathcal{D})$ samples the datapoints corresponding to each fact in the dataset $\mathcal{D}$, grouped by language. $X$ and $Y$ are representations of the same fact in different languages. See Appendix **??** for further details. For our experiments, we let $\mathcal{D}$ be the parallel examples from the training set. $sim_\theta$ is the average cosine similarity between the representations of the two sets of points. The unification score correlates strongly (Pearson's correlation coefficient $>0.95$) with cross-lingual accuracy, for both activation-based and gradient-based representations (Fig 5).

Not only does the unification score correlate strongly with model quality, it can be used to *select* training runs that will generalize well. In fact, as Figure 6 (left) demonstrates, we find that utilizing the unification score can be as effective as collecting a small test set. These experiments were performed over 110 runs varying the fraction of celebrity events (from 0% to 20%), and the ratio of non-celebrity events in the majority vs minority language (from 1:1 to 20:1) and the amount of token overlap within a language. We repeatedly sampled 33% of runs and compared the cross-lingual test performance of the runs chosen by a variety of heuristics. The heuristics we compare are:

**in-Lang Test** : Select the model with the best same-language test performance.

**xLang Test (k%)** : Select the model based on a smaller cross-lingual test set, with k% of the cross-lingual test set being used for model selection. Note that this is a strong baseline, as it effectively 'gives' more data to these methods than other methods.

**Unification Score** : Last token unification score

We also evaluate unification scores as a method for selecting a checkpoint, as it may be expensive or difficult to collect cross-lingual validation data. In Figure 6 (right), we compare to a baseline that selects the last checkpoint among checkpoints with the highest same-language accuracy. While both

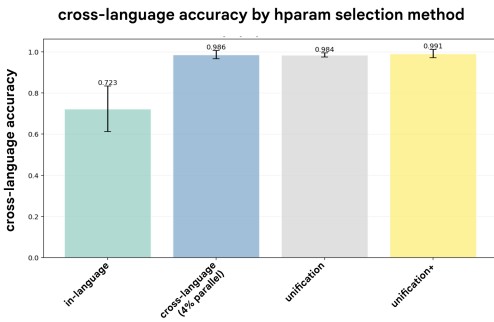
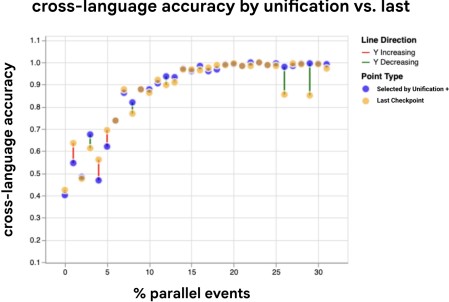

Figure 6: (left) Unification score is competitive with using a small test set to select the best model across multiple hyperparameters. Note that the naive in-language test set selection scheme dramatically underperforms both unification metrics and using a cross-lingual test set. (right) We compare the cross-lingual test performance chosen by different checkpoint selection schemes across runs. Each column represents a different training run with a given fraction of celebrity events. Blue dots indicate cross-lingual test performance of the checkpoint chosen by the unification score, while yellow dots correspond to the last checkpoint with perfect cross-lingual accuracy. A slightly modified version of unification score is used for checkpoint selection, see App. A.17

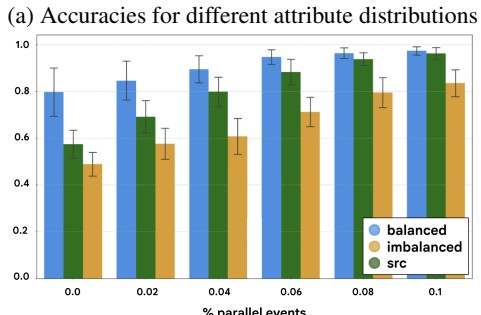
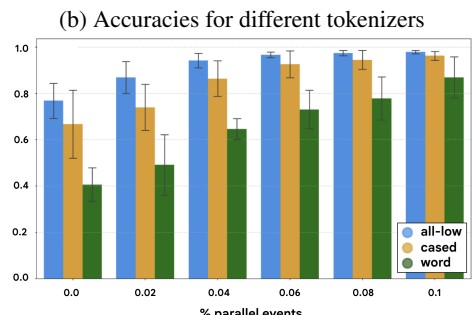

Figure 7: (left) Cross-lingual accuracy for models trained on unaltered, `balanced` and `imbalanced` datasets for different values of parallel events. (right) Cross-lingual accuracy for models trained with the character (`all-low` and `cased`) and `word` tokenizers respectively for different values of parallel events.

methods tend to choose checkpoints with similar quality, selecting using unification scores avoids some scenarios where the model begins overfitting on same-language recall. This represents an exciting new direction for model selection – mechanistic analysis of models can improve our ability to predict generalization. In the next section we corroborate our results with evidence from LLMs.

However, we are also interested in understanding what properties of the dataset cause models to achieve high levels of unification. When do models pay attention to language identity - and why?

## 6 ANALYZING THE LANGUAGE FEATURE

Figure 2a shows both that parallel training improves cross-lingual generalization, and that generalization can take place in the absence of parallel data. As the previous analysis demonstrates, and in line with observations from analyses of large LMs, cross-lingual factual recall fails if the model's internal representations are separated by language. In other words, when *language identity is strongly encoded* in the hidden representations. We hypothesize that the strength of this encoding, or the **language feature footprint**, is determined by the **informativeness** of language identity for the prediction task and its **extractability** from the data. This is theoretically grounded: seminal work by Saxe et al. (2019) shows that models learn features in descending order of the variance they explain in the training data; e.g., a high-level *plant-vs-animal* distinction is learned before more fine-grained categories like *bird-vs-fish*. Similarly, Lampinen et al. (2024) argues that easier-to-learn features tend to dominate a model's internal representations and explain more variance than difficult-to-extract features. To restate our hypothesis:

> If the language identity of an input provides a useful signal about the label (is *informative*) and is easy to recognize (is *extractable*), it will be learned early and dominate the representation. Conversely if language identity is not informative or hard to extract, its representational footprint will be small and representational separation is less likely.

## 6.1 LANGUAGE INFORMATIVENESS

How can the language of an example be a useful signal for prediction? The reason is that while attributes (e.g., *1911*) are strictly speaking a function of the subject entity (e.g., *John Smith*) and the fact type (*birth-year*), the distribution over attribute values is not uniform in our KG or datasets (Fig. 19), thus a spurious correlation exists between the language identity and the predicted attribute. In other words, the mutual information between the language variable and the attribute variable (for a given fact type) is positive. This design mimics real multilingual datasets in that, for example, texts written in Spanish reference Spanish cities more often than texts written in English. The language feature may therefore provide a useful prior over attribute values, helping to reduce loss early in training, before the model learns to recall facts by combining the subject and relation representations (Geva et al., 2023). To test this hypothesis, we create two dataset versions from a base dataset with little (<10%) parallel data and with an equal split between languages (see Appendix A.4 for details):

**balanced** Includes additional examples (corresponding to **new** events) to equalize the example count for every attribute value across the two languages, thereby minimizing the language feature's informativeness.

**imbalanced** Includes the same number of additional examples, created from the same set of new events, but adds them in the language in which the attribute is already present more frequently, amplifying the existing discrepancies and therefore increasing the language feature's informativeness.

Figure 7 shows cross-lingual accuracies for the three settings. Cross-lingual generalization is consistently worst when attribute distributions are imbalanced and mutual information between language and attribute is high (checkerboarding is correspondingly more prevalent in this setting - see Figure 14). In summary, this experiment reveals a clear correlation between the informativeness of the language feature (measured using mutual information) and its footprint in the model's internal representations. These representations, in turn, are strongly predictive of cross-lingual performance.

## 6.2 LANGUAGE EXTRACTABILITY

We next test the hypothesis that making the language feature easier to extract also boosts its influence on the model's representations. In our default setup, the language feature is trivial to extract: our synthetic languages do not share vocabulary, and our tokenizer is word-based. Thus every token (except for entity arguments like names, cities, and years) is a perfect indicator of language identity. To make the language feature harder to extract, we switch to a *character-based* tokenizer while keeping all other aspects of model training unchanged. To isolate the effect of language feature extractability from other potential effects of this tokenizer change, we contrast three settings:

**word** Baseline word-based tokenizer and the original dataset (highly extractable language feature);

**all-low** Char-based tokenizer, with all the templates in lowercase (less extractable);

**cased** Char-based tokenizer, where templates in the first language are in lowercase and templates in the second are uppercase (more easily extractable language feature than `char` but harder than `word` because people's names are spelled normally, introducing lowercase letters).

The last two models are directly comparable, as both use the same character-based tokenizer on essentially the same examples. The only difference is that the language feature is more easily extracted by the latter model, since the token sets used in the two languages are disjoint (upper-case vs lower-case characters). The results in Figure 7 right confirm that when the language feature is more difficult to extract, it has a smaller representational footprint. This has a direct, positive effect on cross-lingual generalization: for the same percentage of parallel data, the `char-all-low` models, where the language feature is least extractable, consistently achieve the highest accuracy.

We observe a similar effect when making language harder to extract by increasing the number of templates. We conduct experiments where there is no token-level indication of language, making it difficult for the model to correctly group all templates in to languages. Increasing the number of templates substantially improves generalization, as demonstrated in Figure 18, while neither increasing the number of events by 10x nor increasing the training duration by 10x improve generalization. We also reproduce these phenomena in a tiny setting, in which a set of one-hot features are directly fed to a logistic regression model with $L_2$ loss. See Appendix A.16 for further details. These experiments suggest a new perspective on the role of parallel data in cross-lingual transfer: A higher proportion of parallel data naturally reduces the information that the language feature contains about attribute distributions.

## 7 DISCUSSION

Our work contributes to the body of research into spurious correlations (Geirhos et al., 2019; McCoy et al., 2019) which loom behind many surprising model failures and generalization challenges (Pearl & Mackenzie, 2018). Similar to recent work by Hermann & Lampinen (2020) and Lampinen et al. (2024), we investigate models' inductive biases and aim to characterize how a feature's complexity and predictivity influence its representation. However, we do so in a setup that imitates factual knowledge acquisition and transfer in LLMs, where feature complexity (i.e., low extractability) and predictivity (i.e., informativeness) emerge naturally from the standard training process. We demonstrate that generalization ability is both indicative of and predicted by shared representations across languages. While previous work (e.g. (Li et al., 2024)) has introduced explicit pretraining strategies for encouraging unified representations, we show that unification can also arise from careful dataset construction. Our findings (Sec. 6) explain why script, shared vocabulary, aligned embeddings and parallel data promote cross-lingual knowledge transfer and is predictive of cross-lingual factual recall (Sec. 2).

Our results suggest two possibilities for improving cross-lingual recall: by obscuring differences between languages, or by balancing attribute frequencies across languages in the pre-training mixture. However language clearly *can* be a useful prior, e.g., in factual queries requiring a language- or culture-specific answer (though how much the model relies on it depends on how easily the language feature can be extracted).

**Limitations** Our synthetic languages are defined solely as sets of templates, thus ignoring structural and lexical (dis)similarities between languages. While this is a clear simplification of LLM pretraining data, our key findings are independent of this design choice.

## 8 CONCLUSIONS

In this work we use a controlled setting to study why LMs often fail at cross-lingual knowledge transfer, hallucinating facts in one language that they know in the other. We demonstrate that these failures are caused by models developing separate, language-specific representations for facts rather than unified, language-agnostic ones. Our key finding is that separation is driven by the informativeness and extractability of the language feature itself and happens very early in training. These results shed light on the role of tokenization, shared vocabulary, script and embedding alignment which have been observed previously but left unexplained. Finally, we introduce a unification score to quantify the representational similarity. This metric is strongly predictive of cross-lingual factual accuracy and can be used for practical model selection.

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

## A  APPENDIX

### A.1  DATA GENERATION PROCEDURE

For readability, Fig. 1 uses English and Spanish templates, but in our setup templates are constructed by randomly sampling tokens from a predefined vocabulary (see App. A.3 for examples). No two

```python
def create_data_splits():
  # generate events
  events = []
  for name in names:
    birth = Event(name, random(cities), random(birth_days))
    death = Event(name, random(cities), random(lifespans) + birth.date)
    events.extend([birth, death])

  # generate templates
  templates["lang0"]["birth", "time"] = []
  for _ in range(N_TEMPLATES):
    tokens = random(tokenizer.vocab(), size=(TEMPLATE_LEN,))
    tokens = randomly_insert(" {subject}", tokens)
    tokens.append(" {time}")
    templates["lang0"]["birth", "time"].append(tokens)
  # same for all other facts (e.g. death place, birth place, ...) and languages

  crosslingual_events, other_events = random_split(events)
  # other events split equally by language

  crosslingual_train_data = cartesian_product(templates, crosslingual_events)
  other_train_data = {
      lang: cartesian_product(templates[lang], other_events[lang])
      for lang in LANGUAGES
  }

  other_train_data, in_distribution_test = drop_equally_by_event(other_train_data)
  if is_too_big(other_train_data):
    other_train_data, extra_in_distribution_test =
        drop_extra_train_data(other_train_data)
    # Note: drop_extra_train_data explicitly maintains the existing ratios
    # of templates per each event in the train set
    in_distribution_test += extra_in_distribution_test

  out_of_distribution_test = {
      lang: cartesian_product(templates − templates[lang], other_events[lang])
      for lang in LANGUAGES
  }
  train_data = other_train_data + crosslingual_train_data
```

Figure 8: Pseudocode for data generation & test set splitting

templates share tokens, so no tokens are shared between languages (except for arguments). In all experiments we use **two languages**, mostly with five templates per language and fact type. See pseudo-code for the data generation process below; sample templates are shown in App. A.3. Since cross-lingual events are seen in (at least) twice as many training examples as the events which are only expressed in a single language, we need to ensure that the total size of the training dataset is invariant to the fraction of parallel data so we can compare models fairly. To this end, we downsample verbalizations of monolingual events. In our experiments the attributes are sampled uniformly, which naturally results in a variability in attribute frequencies (e.g., there may be more people born in *Berlin* than in *Paris*). We have a set of 100 cities (joint between the birth and death events), and two disjoint sets of 20 & 30 years for birth and death years. We also create a custom tokenizer which tokenizes every word as a single token. We include pseudocode for the knowledge-graph generation and partitioning in fig. 8

## A.2 MODEL & TRAINING DETAILS

The models for most of our experiments use the Gemma-2 architecture with models of approximately 2 million parameters. For most experiments, we use the following parameters:

| Hyperparameter | Value |
|---|---|
| Hidden Size ($D_h$) | 128 |
| Intermediate Size ($D_i$) | 512 |
| Head Dimension ($D_{head}$) | 64 |
| Number of Hidden Layers ($L$) | 6 |
| Number of Attention Heads ($H_{attn}$) | 4 |
| Number of Key/Value Heads ($H_{kv}$) | 1 |

We also conduct experiments with Pythia models and with Gemma models with both 0.1x the number of parameters and 10x the number of parameters, but scale did not end up being a substantial variable in this work. Models are trained for 100 epochs (although this is quite gratuitous, they typically converge in a small fraction of this time). We train models with a weight decay of 0.2, although we note that varying this from 0.0 to 0.4 did not change results very much. We train with a batch size of 128 and a cosine learning rate starting at 3e-4.

We include reference training plots in 9.

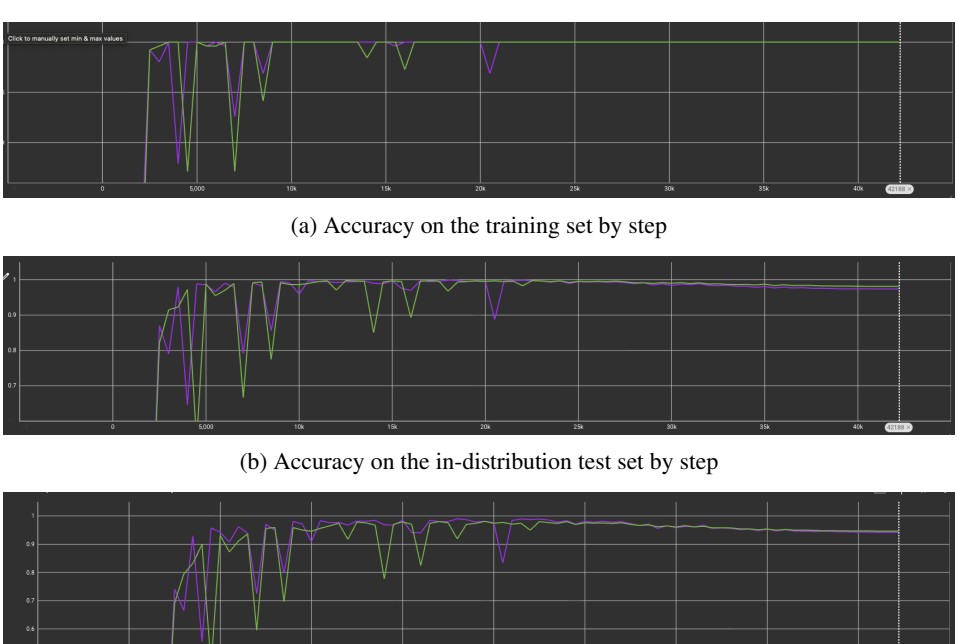

(a) Accuracy on the training set by step

(b) Accuracy on the in-distribution test set by step

(c) Accuracy on the out-of-distribution test set by step

Figure 9: Model performance across various datasets. Subfigures (a), (b), and (c) display results for the training, in-distribution, and out-of-distribution sets, respectively.

```
def compute_unification_scores(model, datapoints):
  total = 0
  for datapoints_by_fact in group_by_fact(datapoints):
    for idx in range(len(datapoints_by_fact)):
      datum = datapoints_by_fact[idx]
      same_lang = [
        d
        for i, d in enumerate(datapoints_by_fact)
        if d.language == datum.language and i != idx
      ]
      other_lang = [
        d
        for i, d in enumerate(datapoints_by_fact)
        if d.language == datum.language and i != idx
      ]
      total += mean_cosine_similarity(model, datum, other_lang) /
        mean_cosine_similarity(model, datum, same_lang)

  return total / len(datapoints)

def mean_cosine_similarity(model, datum, other_data):
  return mean([
    cosine_similarity(
      concat_residual_latents_at_last_token(model, datum),
      concat_residual_latents_at_last_token(model, other_datum)
    )
    for other_datum in other_data
  ])
```

Figure 10: Pseudocode for calculation of unification scores

## A.3 SAMPLE TEMPLATES

| Language Idx | Frame | Template |
|---|---|---|
| 0 | birth | h 56 109 1961 Watkinss {arg0} divorced Mend {arg1} |
| 0 | birth | h 1961 1978 {arg0} When Sp {arg2} |
| 0 | birth | {arg0} Cruzs meet vino Mend 56 When h {arg2} |
| 0 | death | Nguyens {arg0} What Frank Benne house {arg1} |
| 0 | death | house Frank ist {arg0} passed for What W {arg1} |
| 0 | death | {arg0} Frank W Ste ist for {arg2} |
| 1 | birth | {arg0} 1955 concert Schmid occurred deceased Wri finalized {arg1} |
| 1 | birth | {arg0} 1955 Rob Collinss Wheelers deceased 21 {arg2} |
| 1 | birth | Al Wri Palmers {arg0} 1955 Wheelers Pay Schmid {arg1} |
| 1 | death | {arg0} and Thompsons Where that major Hug {arg1} |
| 1 | death | 100 Pal Thompsons 119 {arg0} h {arg2} |
| 1 | death | p 100 lugar Pal and {arg0} Thompsons major {arg1} |

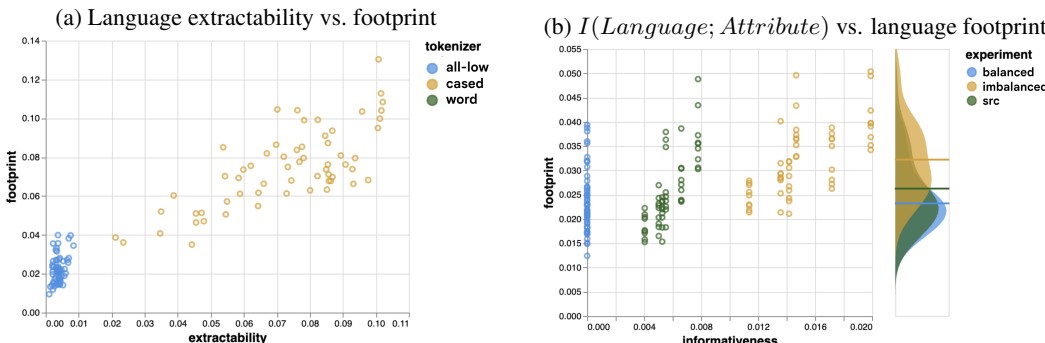

Figure 11: (left) Models' mutual information (MI) between the attribute (label) and the language, plotted against the language feature footprint in the hidden representations (computed as $R^2$). (right) Extractability of the language feature (mutual information between tokens and language), plotted against the language feature footprint in the hidden representations (computed as $R^2$) (only character based results included to enable direct comparison).

### A.4 LANGUAGE INFORMATIVENESS EXPERIMENT DETAILS

These include a language feature, which correlates with (but does not perfectly predict) the output, and a set of features that each uniquely identify an entity. The entity-identifying features are sufficient to perfectly predict the target labels. In this setup we can model *extractability* by independently scaling the language feature and the entity features. As we decrease the magnitude of the language feature relative to the entity features, we observe that train-set accuracy remains fixed at 100% but test set accuracy (measured on previously unseen language-entity pairs) increases towards 100%, supporting our claim that the existence of a confounding variable impedes the creation of the correct circuit (details in Appendix A.16). These results provide a simple lens for understanding the conditions under which a feature irrelevant to the task (such as language identity) can prevent the model from generalizing. Names and attributes are not affected by changes to template casing, i.e., *Alice Brown* and *Paris* are spelled identically in all three settings.

### A.5 UNIFICATION PROBING DETAILS

We note that the correlation between unification & cross-lingual recall accuracy can be highly sensitive to the metric used to quantify unification. As an illustrative example, in 12 we include some plots of an alternative measure that correlates much less strongly with the the chosen measure of generalization.

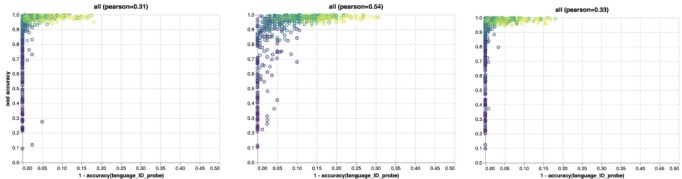

Figure 12: We also experiment with an alternative measure of how similar the representations of celebrities are between languages. Here, we train a linear probe to identify the language that a cross-lingual fact was mentioned in, with the hypothesis that such a probe should fail if the representations of cross-lingual facts are unified (they have the same representation in both languages). We see that this alternative formulation is substantially less discriminative than the one incuded in the main body.

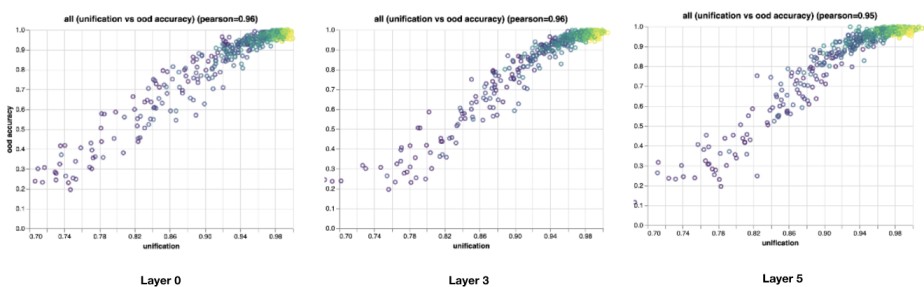

Figure 13: Layerwise results for activation unification scores vs OOD.

## A.6 LANGUAGE CHECKERBOARDING IS MORE OBVIOUS FOR IMBALANCED MODELS

Figure 14: Imbalanced vs. balanced at checkpoint-40,000

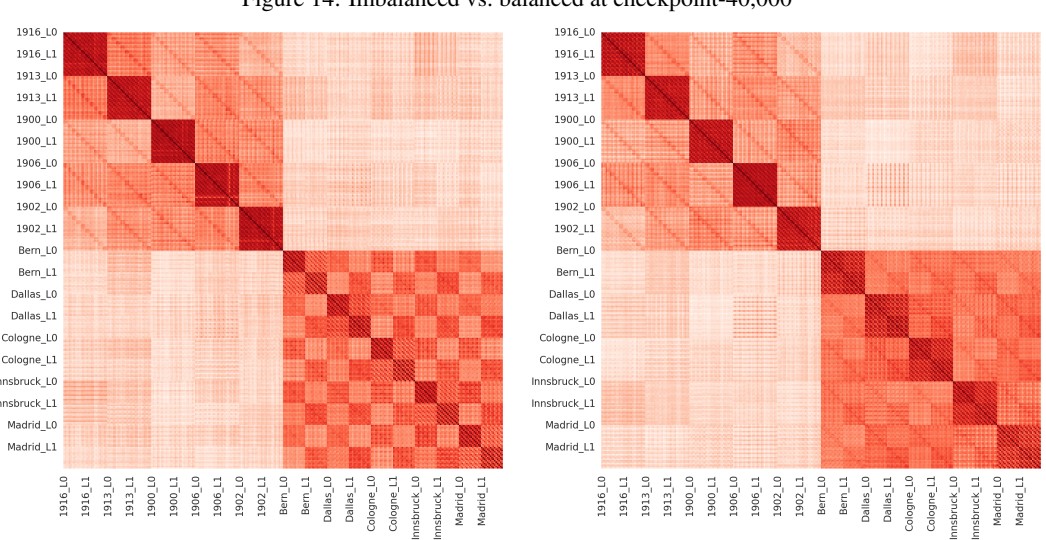

## A.7 LANGUAGE CHECKERBOARDING IS MORE OBVIOUS FOR CASED MODELS

Figure 15: Cased vs all-lowercased (by language) at checkpoint-8,000

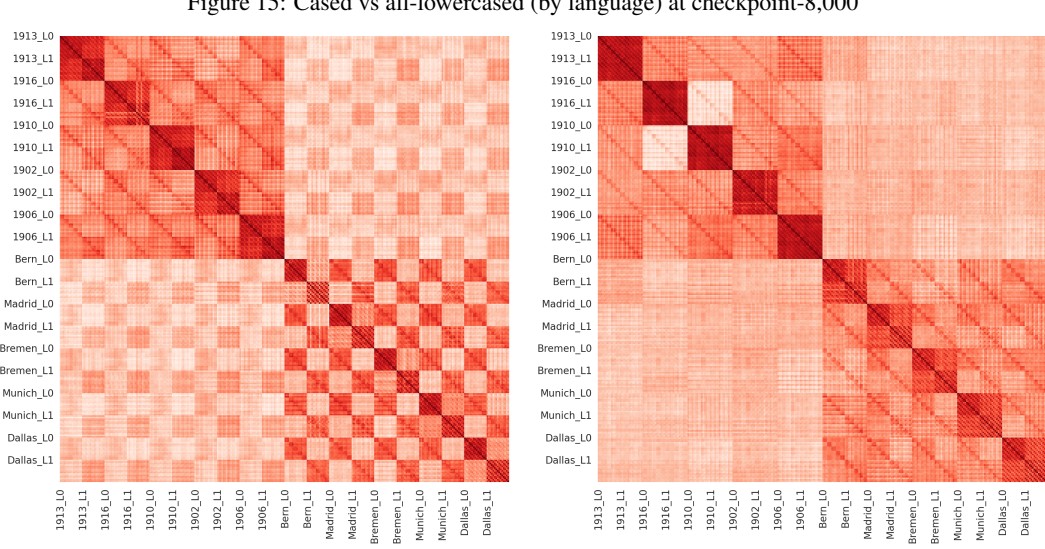

## A.8 LANGUAGE CHECKERBOARDING IS MORE OBVIOUS FOR LOWER LAYERS

Figure 16: [Models with 30% and 8% cross-lingual events, layers 0-1, final checkpoint, activations-based representations] The presence of language checkerboarding (right) is particularly striking in the lowest model layers. For the poorly generalizing model, language identity is the dominant factor determining the similarity of representations for the same fact type. Two birth-city and two death-city attributes (most frequent in the cross-lingual portion of the respective training data) are picked to collect representations.

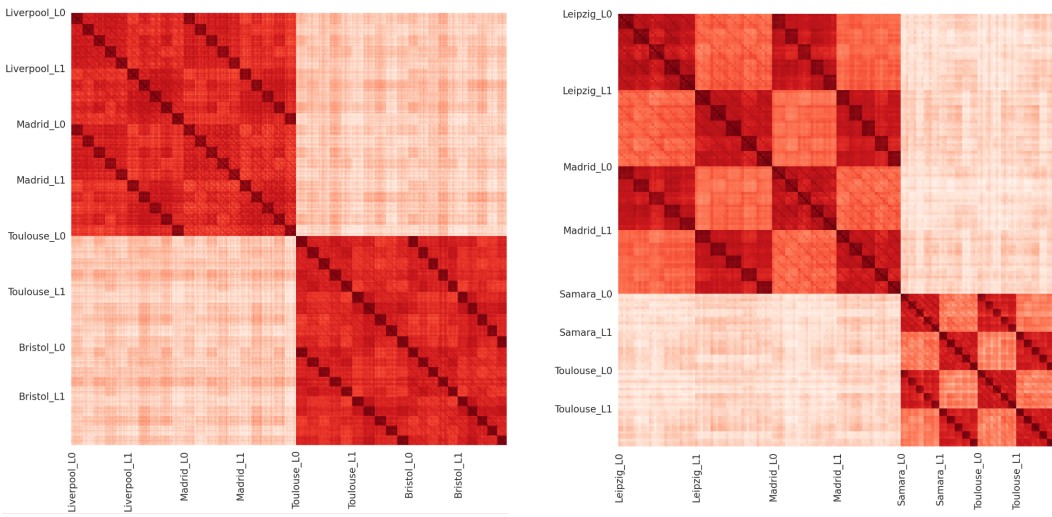

## A.9 GRADIENT-BASED REPRESENTATIONS YIELD SIMILAR PATTERNS

Figure 17: [Models with 30% and 4% cross-lingual events, final checkpoint] Also with **gradient-based** representations the language checkerboarding is visible in a model with poor cross-lingual generalization (right). Three most frequent (in the cross-lingual portion of the respective training data) birth-city attributes) are picked to collect representations.

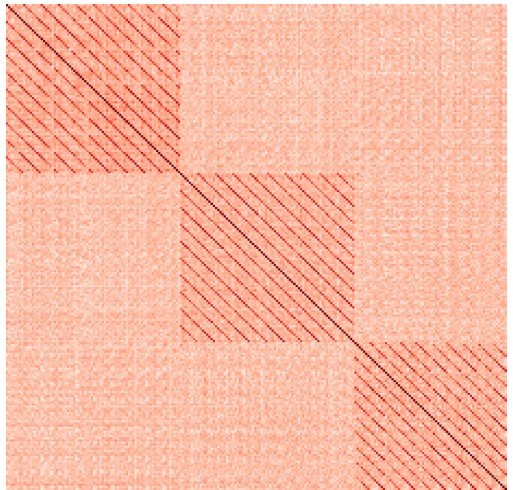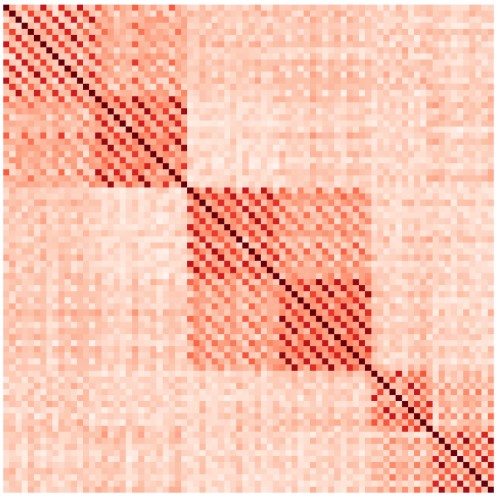

## A.10 ADDING TEMPLATES IMPROVES CROSS-LINGUAL GENERALIZATION

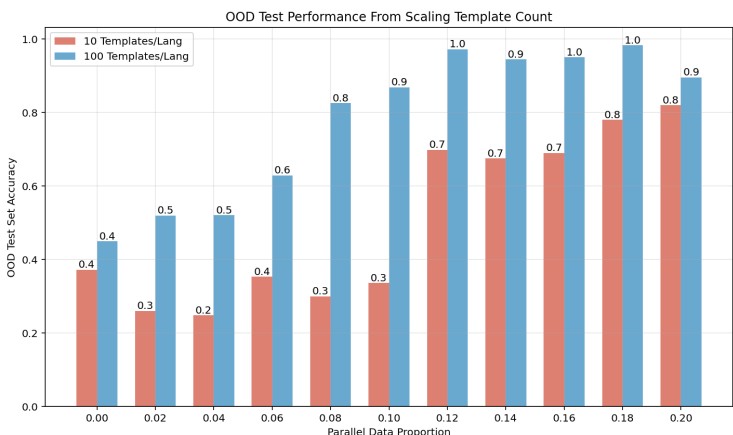

Figure 18: Increasing the number of templates substantially improves cross-lingual generalization, despite also increasing the difficulty of the test set.

## A.11 BALANCED VS IMBALANCED DATASET CONSTRUCTION.

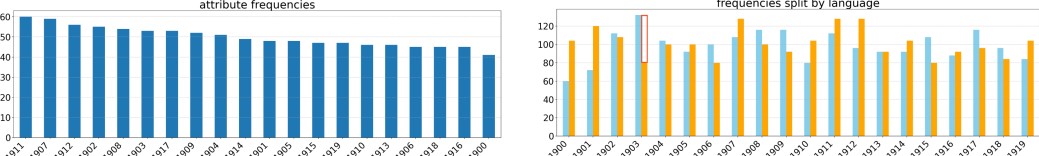

Figure 19: Frequencies of the birth-year attribute values in the KG (left) and in a dataset, split by language (right). The red rectangle highlights the difference in example counts between two languages for a particular year. The red rectangle on the right represents the number of examples for that attribute needing to be added to the second language (orange) to create a balanced dataset, versus being added to the first language (blue) to create an imbalanced dataset.

## A.12 FACT ID VS LANGUAGE ID FEATURE FOOTPRINTS.

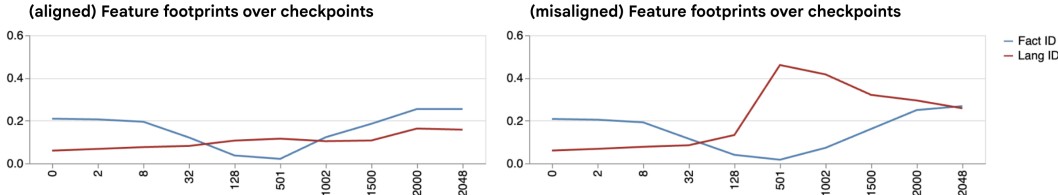

Figure 20: Representational variance explained by language ("Lang ID" - red) versus fact ("Fact ID" - blue) across initial checkpoints. The model on the left is one in which language is uninformative ("Balanced"), whereas language is highly informative for the model on the right ("Imbalanced"). Both models are trained with 0% parallel data, thus language identity is highly extractable. However in the `Imbalanced` model, language identity is also highly informative. The language feature footprint grows quickly in the imbalanced model (before shrinking, although it still ends up larger in imbalanced than in balanced models - see Fig 7).

Following Lampinen et al. (2024), we measure the footprint of a feature as the $R^2$ from the linear regression fit to the representation vectors using the feature values for each training point, i.e. the representational variance explained by that feature. We consider two features: the *distractor* language identity feature and the *true* fact identity feature. The fact identity for an example is a combination of its subject entity and the fact type (examples have the same fact identity if and only if they express the same fact). We observe that where the language identity is highly informative, for

example in the `imbalanced` setting discussed in Section 6.1, the language footprint grows quickly in early checkpoints relative to fact identity (Figure 20).

### A.13   CROSS-LINGUAL GENERALIZATION CAN TAKE PLACE FOR THE WRONG REASONS.

We observe emergence of cross-lingual generalization in celebrities=0 environments, where there is no formal basis for mapping language A templates to language B. (Or, is the better takeaway from the observation that generalization occurs when celebrities=0 that bridge entities are not strictly necessary for cross-lingual transfer? This has been observed.)

### A.14   OVER THE COURSE OF TRAINING, THE MODEL LEARNS, THEN LEARNS TO IGNORE SPURIOUS SIGNALS.

We observe that as training progresses, the fraction of errors explainable by shared name-token confusion rises, then falls.

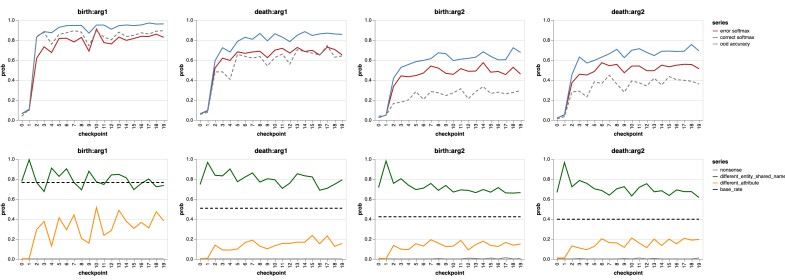

Figure 21: Green line shows proportion of errors where the predicted attributed belongs to a different entity sharing either a first or last name with the test entity, over the course of training for a Pythia model with 0 celebrities.

### A.15   LARGE LM EXPERIMENTS

We validate our findings in multiple open models including Gemma-2-2B (Riviere et al., 2024), Gemma-3-4B(Team et al., 2025), Llama-3.2-3B(Grattafiori et al., 2024), Qwen3-4B (Yang et al., 2025), and Mistral-7B (Jiang et al., 2023), using the ECLeKTic dataset (Goldman et al., 2025), designed to evaluate the cross-lingual transfer ability of large language models. Based on facts which are extracted from Wikipedia articles that exist only in a single language out of twelve, it contains facts in pairs of languages and allows for cross-lingual factual knowledge transfer evaluation. We create additional in-language data by having a LLM (`gemini-2.0-flash`) generate paraphrases. To evaluate, we use the same LLM as an autorater, checking whether provided answers match the correct answer.

**Unification score and explained variance of cross-lingual accuracy**   We leverage the unification metrics from Sec. 5 to analyze cross-lingual generalization within ECLeKTic. In particular, for each fact, we calculate its activation-based unification score and probe whether a higher score is predictive of cross-lingual accuracy. We find that unification predicts the accuracy with an ROC score of around 0.65, regardless of the layer, thus unification provides significant signal regarding model response accuracy. We validate that this is statistically significant at all layers via a t-test with Bonferroni correction (See more details in A.15.5).

**Vocabulary overlap and cross-lingual accuracy**   To replicate our observations regarding the effects of tokenization (Sec. 6.2), we study vocabulary overlap between languages as a predictor of cross-lingual transfer. We follow the approach proposed by Qi et al. (2023) to measure vocabulary similarity. We use Flores-200 (Costa-Jussà et al., 2022) dataset that consists of 2000 sentences translated from English into different languages. We run those sentences through the model tokenizer to get a model-specific vocabulary for each language. For each pair of languages $A$ and $B$ with vocabularies $V_A$ and $V_B$, we calculate their Jaccard similarity ($S = |V_A \cap V_B|/|V_A \cup V_B|$. We then group examples by their source and target language, and for a more intuitive analysis, focus

on examples with the source language of English. We calculate the Pearson correlation between $S$ and cross-lingual accuracy across models and observe an average coefficient of 0.64. This finding aligns with Qi et al. (2023) where they reported a positive correlation between cross-lingual transfer and vocabulary similarity. See per-model details in Appendix A.15.6. Extending the analysis from Sec. 6, the higher the vocabulary overlap, the less extractable the language feature. Therefore the model needs to rely more heavily on semantic information than language information, which should improve cross-lingual generalization ability. Indeed, our analysis (Appendix A.15.6) shows that vocabulary overlap explains a significant amount of variance in cross-lingual accuracy.

In the following sections, we include additional details about the ECLeKTic dataset, our data augmentation phase, autorating, detailed results on the significance of the relationship between unification score and cross-lingual accuracy, and last but not least, results on multiple models highlighting the connection between vocabulary overlap and cross-lignual accuracy.

### A.15.1 Details about the ECLeKTic Dataset

ECLeKTic dataset (Goldman et al., 2025) has been designed to evaluate cross-lingual transfer based on facts that are language-specific. That is, they have a dedicated Wikipedia page in one language, but not in of the other languages covered in the dataset (English, French, German, Hebrew, Hindi, Indonesian, Italian, Japanese, Korean, Mandarin Chinese, Portuguese, and Spanish.) The dataset includes more than 4000 question/answer pairs each addressing a fact known in one language but written in one of the other 11 languages. For more information, see Goldman et al. (2025).

### A.15.2 Data augmentation details

We use `gemini-2.0-flash` to generate five paraphrases for each example, and augment the dataset with each paraphrase as a new example, keeping the target the same. This increases the number of datapoints five times (1,380). For generating paraphrases, we use the following prompt:

```
Consider the following question in this language: SOURCE_LANG,
QUESTION. Please paraphrase this question in the same language
SOURCE_LANG in at least N_PARAPHRASES different ways making sure
that there are no duplicates and the answer remains exactly the
same.  Start each paraphrase with the tag <paraphrase> and end it
with the tag </paraphrase>.  <paraphrase>,
```
where SOURCE_LANG, QUESTION, and N_PARAPHRASES are the corresponding variables.

### A.15.3 Autorating details

Note that for calculating accuracy, we make an autorater by prompting `gemini-2.0-flash` with the following prompt:

```
Consider the following question:  <question> QUESTION </question>
The correct response to this question is <correct_answer>
GROUND_TRUTH </correct_answer>.  A model has generated
the following answer <generated_answer> GENERATED_ANSWER
</generated_answer>.  Is this correct? (yes/no),
```
where QUESTION, GROUND_TRUTH, and GENERATED_ANSWER are the corresponding variables.

### A.15.4 Detailed T-test results

Detailed t-statistics, when comparing unification score across layers between samples with successful vs unsuccessful cross-lingual transfer in `Gemma-2-2B`:

```
L0:  t statistics=1.618, p=0.107
L5:  t statistics=2.885, p=0.004
L10:  t statistics=3.039, p=0.003
L15:  t statistics=3.419, p=0.001
L20:  t statistics=3.368, p=0.001
L25:  t statistics=3.295, p=0.001
```

**L30:  t statistics=3.054, p=0.002**
**L35:  t statistics=2.842, p=0.005**
L40:  t statistics=2.722, p=0.007
L41:  t statistics=1.627, p=0.105

### A.15.5  ROC PLOTS OF UNIFICATION SCORES

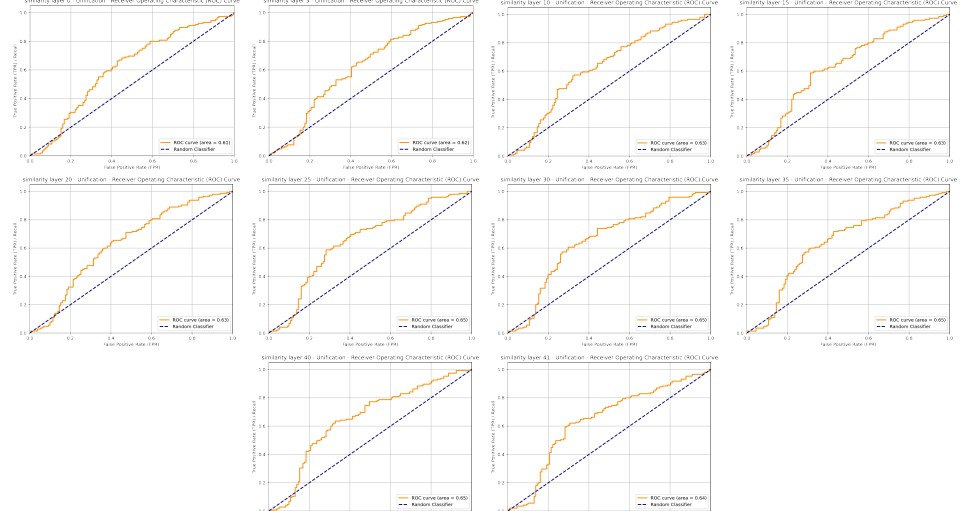

Figure 22: ROC plots of how much unification score is predictive of cross-lingual accuracy for `Gemma-2-2B`.

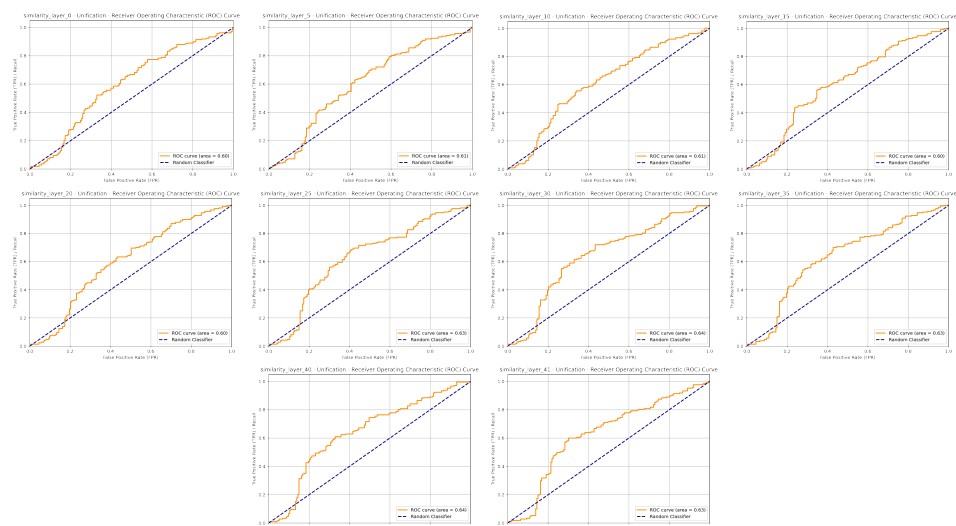

Figure 23: ROC plots of how much unification score is predictive of cross-lingual accuracy for `Gemma-3-4B`.

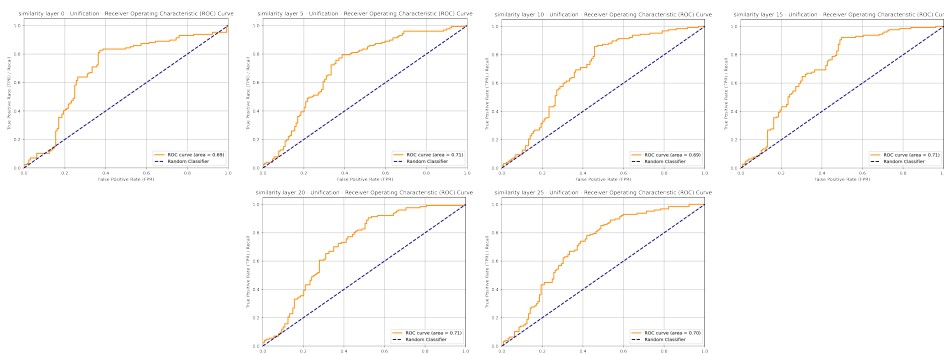

Figure 24: ROC plots of how much unification score is predictive of cross-lingual accuracy for `Llama 3.2 3B Insruct`.

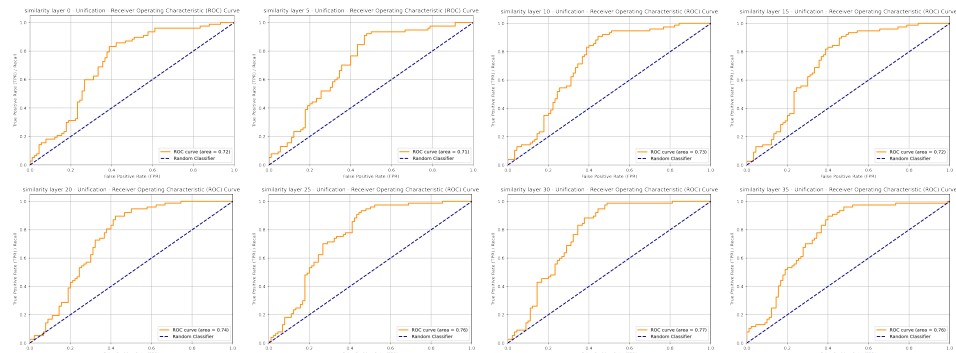

Figure 25: ROC plots of how much unification score is predictive of cross-lingual accuracy for `Qwen 3 4B Instruct`.

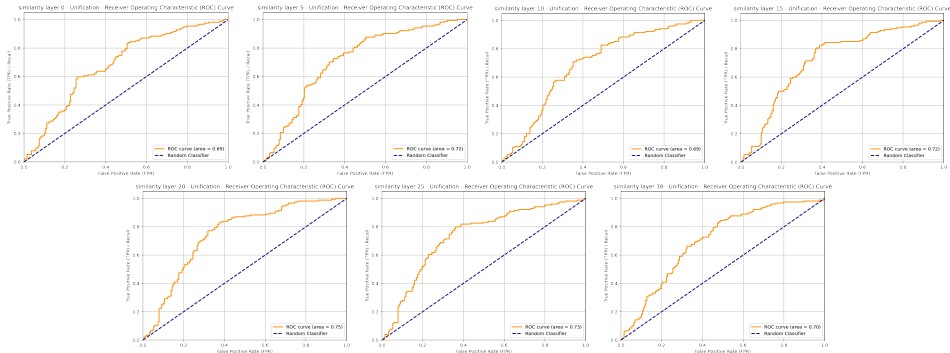

Figure 26: ROC plots of how much unification score is predictive of cross-lingual accuracy for `Mistral 7B Instruct v0.3`.

### A.15.6 VOCAB SIMILARITY VS CROSS-LINGUAL ACCURACY

For a more intuitive analysis, we study the slice of samples where the source language is English (Figure 27). Fore most models, the lower left includes languages like Hebrew that have little to no vocabulary overlap with English. On the other hand, languages such as Indonesian tend to have higher vocabulary overlap and higher cross-lingual accuracy. There is a significant Pearson correlation between cross-lingual accuracy and Jaccard vocabulary similarity. The average correlation coefficient is 0.69, ranging from 0.43 to 0.85 across different models.

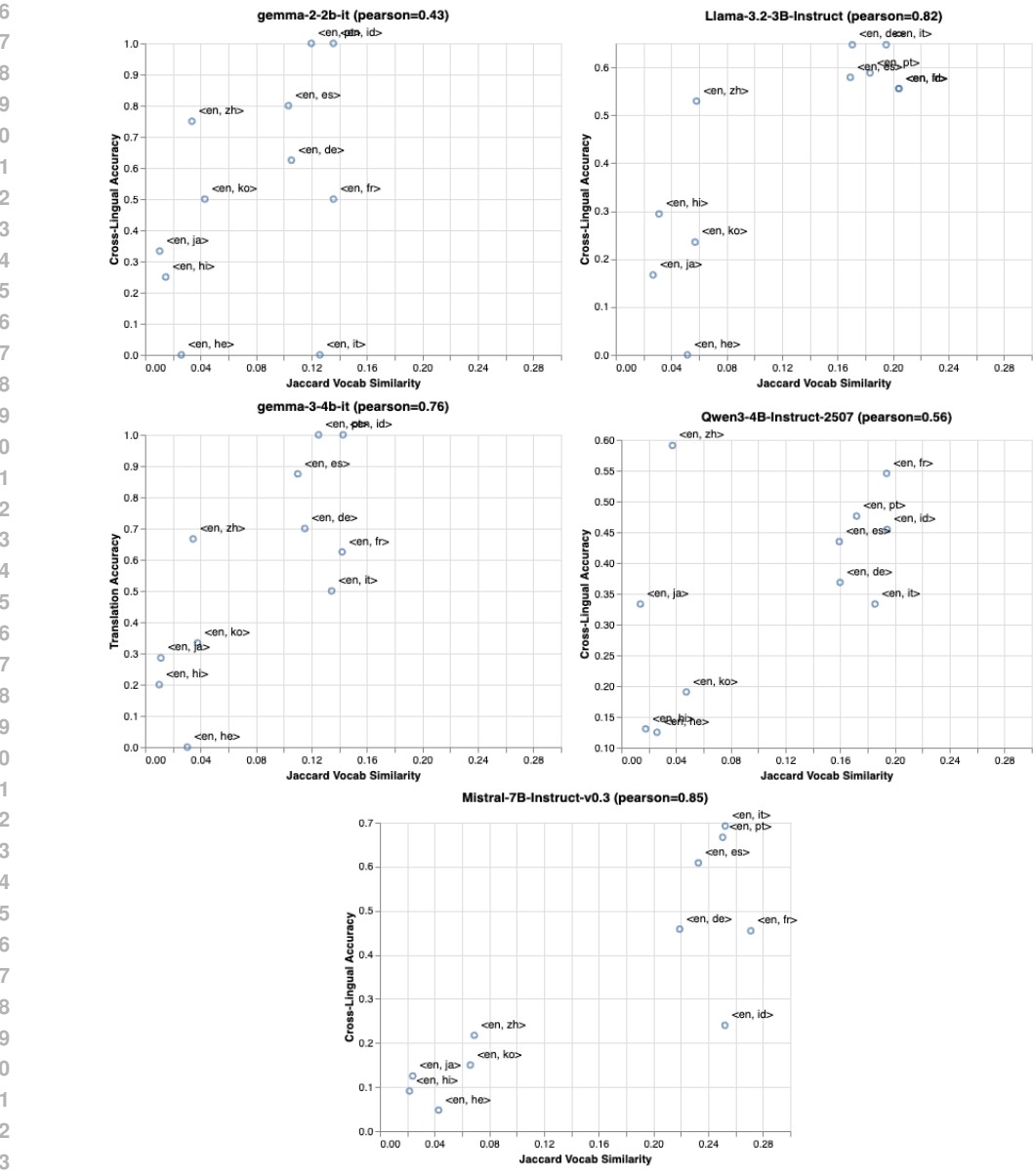

Figure 27: Cross-Lingual accuracy vs. vocabulary similarity. $< x, y >$ indicates going from source language $x$ to target language $y$. There is a significant Pearson correlation between cross-lingual accuracy and Jaccard vocabulary similarity. (0.69 on average)

Table 1: Percentage of variance explained in cross-lingual accuracy when considering vocabulary overlap between source and target languages. Vocabulary overlap explains a good amount of the variance across many source languages.

| Source Language | # Datapoints | $R^2$ (Vocabulary Overlap) |
|---|---|---|
| de | 90 | 68.93 |
| en | 265 | 32.75 |
| es | 95 | 69.06 |
| fr | 45 | 99.53 |
| he | 80 | 31.30 |
| hi | 210 | 41.52 |
| id | 90 | 37.60 |
| it | 120 | 79.99 |
| ja | 140 | 0.03 |
| ko | 65 | 45.94 |
| pt | 95 | 24.14 |
| zh | 85 | 0.20 |

## A.16 UNIFICATION IN REGRESSION SETTINGS

These include a language feature, which correlates with (but does not perfectly predict) the output, and a set of features that each uniquely identify an entity. The entity-identifying features are sufficient to perfectly predict the target labels. In this setup we can model *extractability* by independently scaling the language feature and the entity features. As we decrease the magnitude of the language feature relative to the entity features, we observe that train-set accuracy remains fixed at 100% but test set accuracy (measured on previously unseen language-entity pairs) increases towards 100%, supporting our claim that the existence of a confounding variable impedes the creation of the correct circuit. These results provide a simple lens for understanding the conditions under which a feature irrelevant to the task (such as language identity) can prevent the model from generalizing.

| Language Var. | Same Language Acc (↑) | Cross-Lingual Acc (↑) |
|:---:|:---:|:---:|
| 0% | 100% | 100% |
| 5% | 100% | 77.9% |
| 10% | 100% | 73.3% |

Table 2: Results for model trained in simple regression environment. Language Var represents the fraction of variance in labels explainable by language alone.

## A.17 UNIFICATION SCORE FOR MODEL SELECTION

At random initialization, the unification score is typically around 1.0, because there is no more variation across-languages than there is within language.

As a result of this, we utilize a slightly modified version of the unification score when selecting the best checkpoint. For this, we multiply the unification score with the in-distribution test accuracy. Intuitively, this balances a model that fits the existing data well with one that generalizes well. This metric is listed as 'unification-score+' in the chart.

