# OpenReview forum: "Beyond the Rosetta Stone: Unification Forces in Generalization Dynamics"
_ICLR.cc/2026/Conference — Submitted to ICLR 2026_

### Official Review · Reviewer_w4K3 · 2025-10-31

**Soundness:** 3
**Presentation:** 3
**Contribution:** 2
**Rating:** 4
**Confidence:** 4

**Summary:**

This paper studies why LLMs hallucinate facts when queried across languages, using a synthetic “Petri dish” to train small Transformers from scratch. It finds that models either unify or separate representations of the same fact across languages during an early training phase, and this unification—predicted by a simple similarity-based score—determines cross-lingual generalization.
Crucially, separation is driven by how informative and easily extractable the language identity is;
manipulating monolingual data (balancing attributes, obfuscating language cues) improves transfer without more parallel data. The score also predicts accuracy in Gemma-2B, offering a practical tool for model selection.

**Strengths:**

1.I think the motivation of this paper is solid: why models hallucinate facts in cross-lingual settings, even when they know the fact in another language. This is especially relevant for low-resource languages.
2. The paper is well-written and easy to understand, the experimental design is logically presented, and the figures effectively illustrate key concepts like the unification score and checker-boarding.

**Weaknesses:**

1. I think the baseline is a little weak, only on Gemma-2-2B models, which is not a strong LLM, and without enough justification.
Adding experiments on stronger LLMs such as Qwen, llama, could be justify the performance.
2. While the paper shows that parallel data helps, it does not disentangle whether this is due to increased exposure to facts or reduced language informativeness. A more fine-grained ablation (e.g., parallel data with balanced vs. imbalanced attributes) would clarify this.
3.The paper does not compare its findings to existing multilingual alignment methods (e.g., shared subword vocabularies, alignment losses, code-switching). I would like to see the comparison and analysis against these methods.

**Questions:**

see weakness part.

---

> ### Author Response · Authors · 2025-11-21
> **Thank you for your review**
>
> We appreciate that you found our paper well-written and clear!
>
> **[Weakness 1] We have updated the draft with new results for other models** (Llama-3.2-3B-Instruct, Qwen3-4B-Instruct, and Mistral-7B-Instruct-v0.3, in addition to Gemma 2 and 3 that we already had). **The new results show a high correlation** between the vocabulary overlap and the cross-lingual transfer between a pair of languages, supporting our hypothesis about language extractability and separability. We also merged that section with the related work section (into a new Section 2) to show how our experiments with larger models build on prior work and how all can be explained through the lens of the insights obtained on the tiny scale (namely, the ease of extraction of the language feature).
>
> **[Weakness 2]** We would appreciate further clarification on what you’d like to see. Regarding your request for a “more fine-grained ablation” - **we hope that figure 7a, and the accompanying analysis suffice** to show the effect of parallel data with balanced versus imbalanced attributes. Namely, imbalanced attributes inflate the spurious correlation between language and labels and damage performance. **Is this what you were asking for?**
>
> **[Weakness 3]** As mentioned above, to help highlight our positioning within recent literature, **we have added an explicit comparison with prior work, especially multilingual alignment methods** (e.g. shared subword vocabularies, alignment losses). We omit work on code-switching as an alignment method at the moment but welcome any references that would augment the paper. **Please refer to the updated Section 2:** “Interpreting Existing Observations”. In summary, our lens of language feature extractability contributes a unifying explanation for the efficacy of these seemingly disparate techniques.
>
> If there are any other experiments or clarifications that you would like to see, please let us know. Otherwise, we hope you’ll consider raising your score.

---

### Official Review · Reviewer_XjnT · 2025-11-02

**Soundness:** 3
**Presentation:** 3
**Contribution:** 3
**Rating:** 6
**Confidence:** 4

**Summary:**

The authors present a very interesting study into generalisation in LLMs based on internal representations. Specifically, the authors focus on cross-lingual representations and cross-lingual generalisation. Their study focusses on how specific dataset statistics impact that generalisation. They develop a method of predicting this generalisation based on their developed "unification" score.

**Strengths:**

1. Interesting and detailed analysis into pre-training mixtures and effect on generalisation (specifically in the context of cross-language generalisation)
2. Development of an automatic metric that correlates strongly with task generalisation ("Unification Metric")
3. Development of a synthetic language for such analysis.

**Weaknesses:**

1. Section 3 is quite hard to follow and the plots are very small and hard to make proper use of. (although the general picture of checkboard vs. not checkerboard gets conveyed.
2. Section 6 is very biased to a single model "Gemma" (that is perhaps from the authors themselves [btw, we are not from a "competing" lab, rather this is a scientific assessment]). Secondly, Section 6 is impossible to reproduce from the very short description - limiting it's meaningfulness for the paper as well as a scientific contribution. Section 6 however, is an important contribution to the argument of the paper (as without it synthetic languages form a major limitation).
3. The "Unifcation" metric is probably not sufficiently described to properly reproduce the results.
4. Overall - reproducibility of the paper is limited and it would be hard to verify the results (or their impact).
5. The (presented) results in Section 6 (even though vague) are much lower than on the synthetic data (65% correlation)

**Questions:**

1. Could you describe the Unification metric in more detail. How exactly do you calculate it.
2. Could you describe the KG + Method / Code for producing the synthetic datasets in more detail. Specifically, can you share statistics of your KG and the resulting datasets.
3. Could you expand upon section 6. What are the exact training datasets (+ statistics)? How exactly do you evaluate the model "with LLM judge"? (A follow on would be, how accurate is the method?). etc.
4. Why have you not tried other models (incl. those that are fully open source OLMO, Merlin, GPT-like architectures).

**Details Of Ethics Concerns:**

Not a big concern, however, the authors clearly present work about Gemma and analysis of Gemma only. Indicating a strong bias towards the LAB's own work and sharing it openly.

---

> ### Author Response · Authors · 2025-11-21
> **Thank you for your review**
>
> We believe that your comments and clarification questions all improve the presentation and reproducibility of the draft and were easily addressed (see below). We have updated the draft with larger plots (in both the main text and the Appendix), added more details about the dataset and the algorithm, improved the presentation of the unification score, and also extended experiments to other open models.
>
> 1. The intuition behind the unification score is to **identify whether the representations for each fact are relatively clustered by language**. To do so, we use cosine similarity between representations as our similarity metric, deriving representations by concatenating the latents in the residual stream across all layers for the last token. For a more detailed explanation, **we have included pseudocode in the appendix**, titled “Pseudocode for calculation of unification scores”.
> 2. We include **pseudocode for the KG graph generation in the appendix**, titled “Pseudocode for data generation & test set splitting”. Our experiments include 4 types of facts (birth-city, death-city, birth-year, death-year). We conduct experiments at a variety of scales ranging from 10 templates per language to 100 and ranging from 2K facts per type to 20K facts per type.
> 3. **For the LLM experiments, no additional training is conducted**. The ECLeKTic Dataset (Goldman et al., 2025) is used on off-the-shelf models purely for evaluation purposes. The only modification is augmenting the dataset by including 5 paraphrases of each sample. **Appendix A.15 provides details** on how this augmentation is done (A.15.2) and how the answers are autorated (A.15.3). We have also added another section to the appendix summarizing the ECLeKTic dataset statistics (A.15.1).
> 4. **In addition to Gemma 2 2B and Gemma 3 4B, we have conducted experiments on Llama-3.2-3B-Instruct, Qwen3-4B-Instruct, and Mistral-7B-Instruct-v0.3** and included them in the **new Section 2 of the updated draft**. In that new section, we show how prior observations concerning cross-lingual knowledge transfer and our experimental results with LLMs can be explained through the lens of the insights obtained on the tiny scale (namely, the ease of extraction of the language feature). The new results we added show a high correlation between the vocabulary overlap and the cross-lingual transfer between a pair of languages, supporting our hypothesis about language extractability and separability. We have also **updated Appendix A.15 with all the new experimental results**.

---

### Official Review · Reviewer_41AX · 2025-11-03

**Soundness:** 2
**Presentation:** 2
**Contribution:** 2
**Rating:** 4
**Confidence:** 3

**Summary:**

This paper studies the generalization of knowledge across languages. The authors suggest a Petri dish setup, where synthetic knowledge graphs are used to sample bilingual datasets, to control shared information across languages, and then train a tiny transformer to observe the generalization across languages from multiple checkpoints during training. To facilitate the observation, the authors consider Unification Score, a metric to evaluate the difference between representations in two languages. Finally, the authors leverage Unification Score to examine Gemma-2B on the ECLeKTic dataset (Goldman et al., 2025), attempting to verify all findings for a  real LLM.

**Strengths:**

Understanding and interpreting the emergence of cross-lingual generalization is an interesting question as it helps understand how to effectively train an LLM to support multiple languages, esp. low-resource languages.

**Weaknesses:**

1. The presentation of this paper is not very good, making this paper vague, unclear, and verbose. For example, in line 258, the authors state “Concretely, the unification score captures the similarity between semantically equivalent cross-lingual datapoints against a baseline of similarity between semantically distinct same-language datapoints.” , but in the following equation, they define “sim(x, y) against sim (x, x)”. Also, I believe including statistics of the synthetic datasets and more experimental setups (e.g., training steps, batch size, and learning curves) could strengthen the paper.

2. The main finding of this paper is that explosibility or frequency of appearance is the key cross-lingual transfer. However, there is a body of studies focusing on this idea. For example, [1] set up a similar experiment by controlling parallel datasets.

3. The last part, 6 LARGE LM EXPERIMENTS, is not convincing and seems disconnected from other findings. There is an important confounding factor that the frequency of appearance is not the same across languages. For example, a word X appears in English 10k times, but 1 time in other languages, thus it will increase the Jaccard similarity but not improve cross-lingual transfer or predict the cross-lingual transfer, according to other experiments in this paper.

[1] Cross-Lingual Transfer of Cultural Knowledge, ACL 2025

**Questions:**

Refer to Weaknesses

---

> ### Author Response · Authors · 2025-11-21
> **Thank you for your review**
>
> There seems to be a considerable misunderstanding of what we actually propose and what we find. The main criticism seems to concern novelty, and specifically that our “main finding” is that “frequency of appearance is the key cross-lingual transfer”, as several other papers have pointed out. We would like to emphasize that this is *not* our finding, as we **control for frequency and show that, while it may be a factor, cross-lingual transfer may succeed and fail even if the frequency (of tokens, or of facts) is uniform**. We show that the key factors are are language extractability and the mutual information between factual attributes and the language identity.
>
> We have updated the draft to clearly list information about the training process in appendix A.2 “Model & Training Details”. Pseudocode for computing the unification score is listed in the appendix as well, under “Pseudocode for calculation of unification score”.
>
> We hope these clarifications resolve the confusion regarding the main contributions of this work. As the major concerns raised in the review stemmed from this interpretation, we hope you will reassess the paper and raise your rating to reflect the clarification.

---

### Author Response · Authors · 2025-11-21
**Thank you to all reviewers!**

We appreciate the interest in the details around the unification score, knowledge graph generation and data. To this end, we’ve provided pseudocode for these in the appendix (A.1, A.2, A.3, A.16, A.17).

Regarding how the findings relate to large LMs, we updated the last two sections to clearly articulate the link from the tiny-scale experiments to LLMs and added results for three more open models (Llama-3.2-3B-Instruct, Qwen3-4B-Instruct, and Mistral-7B-Instruct-v0.3, in addition to Gemma 2 and Gemma 3 that we had) (A.15.5: Figs.22-26; A.15.6: Fig.27). Experimental results on larger LMs (both ours and from prior work) are provided to confirm that our findings are not specific to our petri dish environment. Rather, they provide an explanation for why shared script and vocabulary overlap correlate with cross-lingual performance, and why techniques like embedding alignment promote cross-lingual generalization. Namely, the representational footprint of the language feature depends on its extractability early in training, and a larger footprint causes the separation of semantically equivalent factual statements.

---

### Meta-Review · Area_Chair_56ic · 2026-01-13

**Summary:**

This paper studies how and why LLMs tend to hallucinate facts when prompted across languages. The authors refer to their approach as a "Petri dish" methodology, which refers to the standard technique of running carefully controlled synthetic toy experiments. The hypothesis is that cross-lingual generalization is driven by the dynamics of representation learning during training. Reviewers generally appreciated the idea, but limited evaluation, lack of clarity and limited enthusiasm from reviewers make it difficult to recommend for acceptance.

**Reviewer Concerns:**

- 41AX presentation: N/A
- 41AX related work: N/A
- 41AX large experiments: N/A
- XjnT sec 3: **No**
- XjnT gemma bias: Yes
- XjnT reproducibility: **No**
- XjnT performance drop on real data: **No**
- w4K3 weak baseline: Yes
- w4K3 better ablation: **No**
- w4K3 comparison to existing methods: **No**

**Reviewer Scores:**

41AX 4->4
XjnT 6->6
w4K3 4->4

---

### Decision · Program_Chairs · 2026-01-26

Reject